# Executive Functions and Deafness: Results in a Group of Cochlear Implanted Children

**Andrea De Giacomo** [1], **Alessandra Murri** [2,*], **Emilia Matera** [1], **Francesco Pompamea** [1], **Francesco Craig** [1], **Francesca Giagnotti** [2], **Roberto Bartoli** [2] and **Nicola Quaranta** [2]

1    Child Neuropsychiatry Unit, Department of Biomedical Sciences, Neurosciences and Sense Organs, University of Bari, 70125 Bari, Italy; andrea.degiacomo@uniba.it (A.D.G.); emilia.matera@uniba.it (E.M.); fpompamea@gmail.com (F.P.); francesco.craig@uniba.it (F.C.)
2    ENT Clinic, Department of Biomedical Sciences, Neurosciences and Sense Organs, University of Bari, 70125 Bari, Italy; francesca.giagnotti@policlinico.ba.it (F.G.); roberto.bartoli@policlinico.ba.it (R.B.); nicolaantonioadolfo.quaranta@uniba.it (N.Q.)
*    Correspondence: alessandra_murri@libero.it; Tel.: +39-080-5592-387

**Abstract:** Objects: This study aimed to evaluate the Executive Function (EF) domains in a group of profoundly deaf children treated with cochlear implant (CI) in comparison to normal hearing (NH) children. The secondary aim was to evaluate the influence exerted by the age at cochlear implant activation on EFs. Materials and Methods: 32 children were enrolled into two groups: group A of 17 CI users with a mean age of 8.78 years and group B of 15 NH subjects with a mean age of 7.99 years (SD + 2.3). All subjects were tested using the following tests: the subtests for working memory of the neuropsychological evaluation battery for the developmental age (Batteria di valutazione neuropsicologica per l'età evolutive), inhibition and control of the impulsive response—CAF, and the tower of London test. Results: No children with CIs scored within the normal range in the tests administered for the evaluation of EF domains. The same scores were significantly lower when compared with scores obtained by NH children. Children with younger age at CI activation showed better executive performances in planning, working memory (backward digit span), and cognitive flexibility (categorical fluency). Conclusion: The results of this study highlight that cochlear implantation plays a role in improving hearing and consequently influences the development of EFs in deaf children.

**Keywords:** cochlear implant; child; executive functions; working memory

## 1. Introduction

Childhood severe-to-profound sensorineural hearing loss (SSHL) is a serious worldwide health problem since it compromises the normal development of verbal language, social integration, and school learning. In Italy, about 1 in 1000 children are born with a hearing deficit.

During the 1990s, cochlear implantation became available as a medical treatment for children with SSHL. The electric stimulation provided by cochlear implants (CIs) has enabled speech perception and the development of spoken language skills in many severe to profoundly deaf children [1–3]. The CI has crucial importance in stimulating the auditory cortex which, after a period of auditory isolation, begins to detect sensory stimuli.

The brain is a highly interconnected information-processing system that develops based on complex interactions between neural activity and sensory stimulation from the environment, including auditory stimulation. As a result, deprivation in early auditory experiences influences the development of other neurobiological and cognitive functions extending beyond spoken language skills [3,4]. Early auditory experiences provide temporal patterns to the developing brain, which may be important for the development of sequential processing abilities such as sustained attention and memory for serially

presented items [5]. Sustained attention and sequential memory processes are critical developmental building blocks for executive functions (EFs) a large and complex group of high neurocognitive processes involved in the search for strategies to achieve goals and modify/monitor behaviors based on environmental variations [6]. EFs are defined as an "umbrella construct" which includes several cognitive processes such as controlled attention, planning, working memory, cognitive flexibility, and inhibition/impulse control processes [7].

A large body of research has demonstrated that EFs and language are dependent on each other for development, particularly through childhood [8,9] and because they are part of an information-processing system and interconnect at various levels in the cerebral cortex, both are affected by the period of auditory deprivation in children with HL.

This study aimed to evaluate the EF domains, including working memory, cognitive flexibility, and planning in a group of children affected by prelingual SSHL and treated with CI in comparison to normal hearing (NH) children. The secondary aim was to evaluate the influence exerted by the age at CI activation on EFs.

## 2. Materials and Method

### 2.1. Ethical Considerations

All procedures contributing to this work comply with the ethical standards of the relevant national and institutional guidelines on human experimentation and with the Helsinki Declaration of 1975, as revised in 2008. This research received no specific grant from any funding agency or the commercial or not-for-profit sectors.

The study design and the subjects' recruitment were approved by the Institutional Review Board Statement of Hospital Consortium Policlinico of Bari, Italy, and were conducted according with the ethical standards of Hospital Consortium Policlinico of Bari, Italy. This research received the approval of the Ethics Committee of Hospital Consortium Policlinico of Bari, Italy (approval number: 3845, protocol code 264/CE).

The recruited families gave written informed consent for the assessment of their child before commencing any study-related procedure.

### 2.2. Subjects

A total of 32 subjects were enrolled into two groups: group A consisted of children affected by prelingual SSHL treated with CI and group B consisted of NH children recruited from a primary school.

Group A: the sample consisted of 17 CI users (7 females, 10 males) with a mean age of 8.78 years (SD ± 2.69). The mean age at CI was 2.03 years. A total of 14 participants had unilateral CI and 3 had bilateral CIs (Table 1). SSHL was caused by Connexin 26 mutation in 10 children and was of unknown etiology in the remaining 7.

**Table 1.** Participant demographic characteristics of Group A—cochlear implant children.

| Subject | Gender | Age at Executive Functions Evaluation (Year) | Age at Cochlear Implant Activation (Year) | Hearing Age (Year) |
|---------|--------|----------------------------------------------|-------------------------------------------|--------------------|
| 1 | M | 6.02 | 1.04 | 4.08 |
| 2 | F | 9.11 | 1.07 | 8.04 |
| 3 | M | 7.1 | 1.09 (Right ear) | 6.01 (Right ear) |
|   |   |   | 4.04 (Left ear) | 3.06 (Left ear) |
| 4 | M | 9.11 | 1.07 | 8.04 |
| 5 | F | 5.11 | 2.04 (simultaneous bilateral) | 3.07 |
| 6 | M | 7.05 | 1.07 | 5.1 |
| 7 | F | 6.07 | 1.05 | 5.02 |

**Table 1.** *Cont.*

| Subject | Gender | Age at Executive Functions Evaluation (Year) | Age at Cochlear Implant Activation (Year) | Hearing Age (Year) |
|---|---|---|---|---|
| 8 | M | 5.06 | 1.09 (simultaneous bilateral) | 3.09 |
| 9 | M | 12.06 | 5.02 | 7.04 |
| 10 | F | 8.11 | 2.08 | 6.03 |
| 11 | M | 7.11 | 1.09 | 6.02 |
| 12 | F | 12.08 | 3.07 | 9.01 |
| 13 | M | 11.1 | 2.02 | 9.08 |
| 14 | F | 11.03 | 7.05 | 4.08 |
| 15 | M | 5.06 | 2.02 | 3.04 |
| 16 | F | 12 | 3.11 | 8.01 |
| 17 | M | 11.05 | 1.03 | 10.02 |
| Mean | | 8.48411765 | 2.27071429 | 5.9911 |
| SD | | 2.6398936 | 1.80104442 | 2.32319731 |

The enrolled subjects fulfilled the following inclusion criteria:

(a)  a diagnosis of prelingual SSHL before the age of 3 years;
(b)  cochlear implantation before 7 years of age;
(c)  at least 4 years of CI use at the time of testing;
(d)  consistent use of a currently available, state-of-the-art multichannel CI system;
(e)  Italian as primary language;
(f)  auditory-verbal rehabilitation with the development of language skills in comprehension and production.

Group B: the NH group consisted of 15 healthy subjects (5 females, 10 males) with a mean age of 7.99 years (SD + 2.3). All children had a nonverbal IQ score within 1 standard deviation of the norm mean or higher, and passed a basic audiometric hearing screening assessment (each ear was tested individually with headphones at frequencies of 500, 1000, 2000, and 4000 Hz at 20 dB).

Potential CI participants and NH children were excluded if (a) a comorbid developmental or neurocognitive delay or disability other than the hearing loss was indicated by the medical chart or parental report or (b) their nonverbal IQ score was greater than 1 standard deviation below the normative mean.

*2.3. Procedures*

For both groups and during a single session, a team of physicians (child neuropsychiatrist) and speech therapists collected clinical and sociodemographic data and administered the following standardized instruments:

Neuropsychological evaluation battery for the developmental age (*Batteria di valutazione neuropsicologica per l'età evolutiva*) (BVN 5–11, BVN 12–18) [10,11]. This is a battery of neuropsychological tests for a complete analysis of high neurocognitive functions including language, visual perception, working memory, attention, reading, writing, and calculating in children aged 5 to 11 and 12 to 18 years.

Working memory (WM) was evaluated with two subtests, the forward and backward digit span. They consisted of an immediate, direct, and reverse repetition of a number series; the span is obtained by proposing a series of increasing length of linguistic items and evaluating which is the longest series that the child can repeat correctly, also observing the order in which the numbers were presented. Each series consisted of 3 items and was considered passed when at least two out of three items were repeated correctly. The administration stopped when the child failed two out of three items.

Phonemic fluency and categorical fluency subtests were used to evaluate cognitive flexibility skills. In the phonemic fluency subtest, it is required to produce many words as possible starting with a specific phoneme (/k/, /s/, and /p/). In the categorical fluency subtest, it is required to produce words semantically associated with each other. Both subtests are timed: the child is allowed one minute to recall the words. The score is given by the number of correct words produced in that time.

*Inhibition And Control Of The Impulsive Response-CAF (6–12 Years)* [12]: the CAF derives from the Junior Hayling Test by Shallice et al. and evaluates the inhibitory capacity, that is, the ability to block the spontaneous response to which is added the ability to produce an acceptable alternative. The test consists of 20 sentences, divided into two groups, in which the final word is missing; the child must perform two different tasks alternating with each other: he must complete a group of sentences with the missing word, while he must inhibit the correct response in the other group and provide a response that is not semantically related to the stimulus and the correct response. The child is asked to alternate the sentences to be completed and to be inhibited, requiring greater flexibility in carrying out the test; with a hand signal, the examiner indicates which sentences to complete with an alternative word. In the evaluation of the CAF, a score is assigned for each sentence to be completed inconsistently with the stimulus; the calculated score is an error score. Three types of answers are highlighted: sentence completions with the missed word demonstrating that they have not respected the delivery (answers C), semantically connected words (answers S), words not semantically connected but not strategic (answers U), and finally words not semantically linked in which there is the use of a strategy (US responses).

*Tower of London* (TOL) [13]: this consists of 12 subtests of gradual difficulty to evaluate planning skills. Always starting from a basic position, the child must follow a sequence of movements to obtain a certain configuration with color balls inserted in specific support. During the execution of each item, only one ball is allowed to be moved at a time from one stick to another (one ball on the small stick, two on the medium stick, three on the large stick), through a restricted number of movements. It is not allowed to put a ball on the table or have more than one in a hand. In recording the score, the total score of the correct answers, the number of violated rules, and the time are considered. The total score of the correct answers measures the ability to plan and monitor actions until the goal is achieved; the number of violated rules measures the ability to understand and keep in mind the presented rules. In the evaluation of the time, the decision time, the execution time, and the total time are used; the decision time is calculated from the presentation of the model to the moment in which the first ball is extracted from the stick, the execution time is calculated from the beginning of the first movement to the end of the final movement of an attempt, finally, the total time is given by the sum of the two previous times.

All the enrolled children acquired verbal language; therefore, all the instructions were given orally although all EF tests had minimal auditory requirements. When necessary, spoken directions were supplemented with nonverbal demonstrations to ensure that all children fully understood the tasks.

### 2.4. Statistical Analysis

All the variables studied were subjected to statistical analysis. Descriptive analysis was carried out for the socio-demographic and clinical features of the group. To study EFs variables the nonparametric test Mann–Whitney was used for assessing whether one of two samples of independent observations tended to have larger values than the other. Spearman's non-parametric rho test was used to evaluate whether there were significant correlations between EFs (working memory, planning, cognitive flexibility) and age at CI activation in group A. The significance level was set at a *P* value less than 0.05. For statistical processing, we used the data processing program Statistical Package for Social Science, version 20.0 (IBM Corporation, New Orchard Road Armonk, New York, NY, USA).

## 3. Results

Group A consisted of 17 children (7 females, 10 males) with a mean age of 8.78 years (SD + 2.69) and a mean age at CI activation of 2.03 years. A total of 14 participants of group A had monolateral CIs and 3 had bilateral CIs. The daily use of speech processors was assessed through data logging and it was more than 10 h a day. All CI children scored on the Category of Auditory Performance-2 (CAP-2) [14] from 5 (understanding of common phrases without lip reading) to 7 (use of telephone with known listener). At evaluation age, they had a receptive vocabulary mean raw score of 72.7 (SD ± 15.6) on the Peabody Picture Vocabulary Test, [15] Italian edition [16]. Morphosyntactic comprehension assessment was undertaken using the Italian version of the Test for Reception of Grammar (TROG)-2 [17]. Group A scored =/> 1 standard deviation from mean.

Group B consisted of 15 NH healthy subjects (5 females, 10 males) with a mean age of 7.99 years (SD + 2.3).

The CI (group A) and NH (Group B) samples did not differ in nonverbal IQ, age, family income, race, or gender.

The scores of Group A were compared to Group B of NH children (Table 2).

**Table 2.** Executive Function subtests results in Group A and B.

| | Group A: Cochlear Implant Children, *n* = 17 (Mean Score ± Standard Deviation) | Group B: Normal Hearing Children, *n* = 14 (Mean Score ± Standard Deviation) | Z | *p* |
|---|---|---|---|---|
| Age at Evaluation | 8.78 ± 2.69 | 7.99 ± 2.3 | −1.173 | 0.246 |
| Forward Digit Span | 3.53 ± 1.28 | 5.29 ± 0.82 | −3.765 | 0 |
| Backward Digit Span | 2.53 ± 0.943 | 3.64 ± 1.33 | −2.303 | 0.026 |
| Inhibition and Control of the Impulsive Response-CAF | 16.59 ± 5.84 | 8.57 + 8.89 | −2.767 | 0.005 |
| Phonemic Fluency | 14.00 ± 10.02 | 28.93 ± 13.53 | −3.021 | 0.002 |
| Categorical Fluency | 34.00 ± 14 | 57.07 + 18.42 | −3.157 | 0.001 |
| TOL Total Correct Score | 22.59 ± 5.85 | 30.43 ± 2.53 | −3.714 | 0 |
| TOL Violated Rules | 3.18 ± 2.96 | 0.00 | −4.3 | 0 |

### 3.1. Working Memory

WM was evaluated with two subtests, the forward and backward digit span. Group A had a forward digit span mean score of 3.53 ± 1.28 and a backward digit span mean score of 2.53 ± 0.943. NH children (group B) performed better at both subtests, having a forward digit span mean score of 5.29 ± 0.82 and a backward digit span mean score of 3.64 ± 1.33.

A statistically significant difference of forward digit span mean score (z = −3.76, *p* = 0.00) and backward digit span (z = −2.30, *p* = 0.02) mean score between group A and B were found (see Figures 1 and 2).

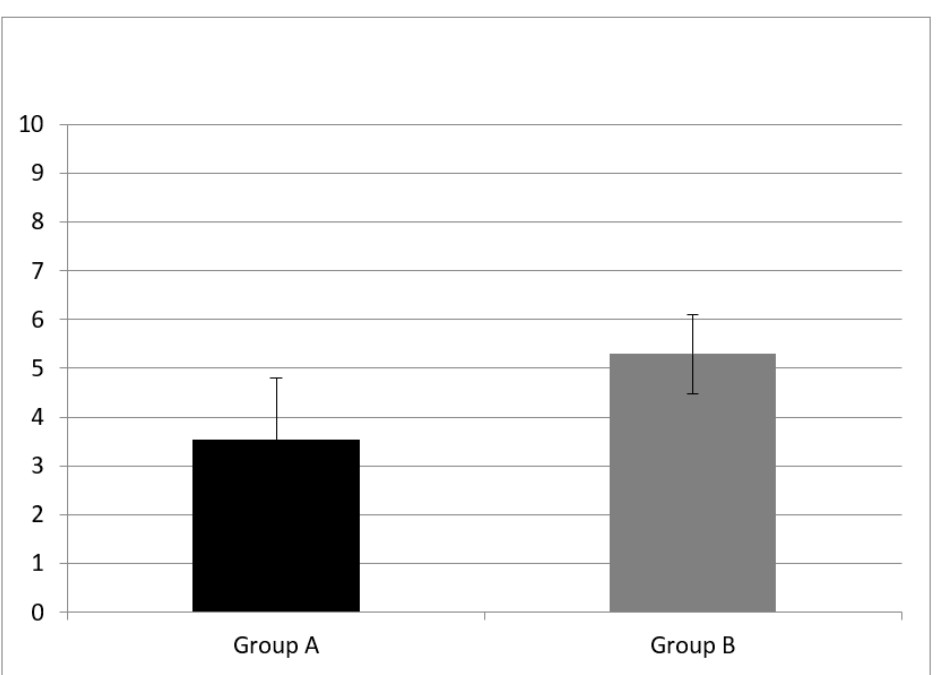

**Figure 1.** Working memory results at subtest of forward digit span in Group A and B. CI children (Group A) had a forward digit span mean score of 3.53 ± 1.28; NH peers (Group B) had mean score of 5.29 ± 0.82. Error bars indicate the standard deviation. A statistical difference was found ($z = -3.76$, $p = 0.00$).

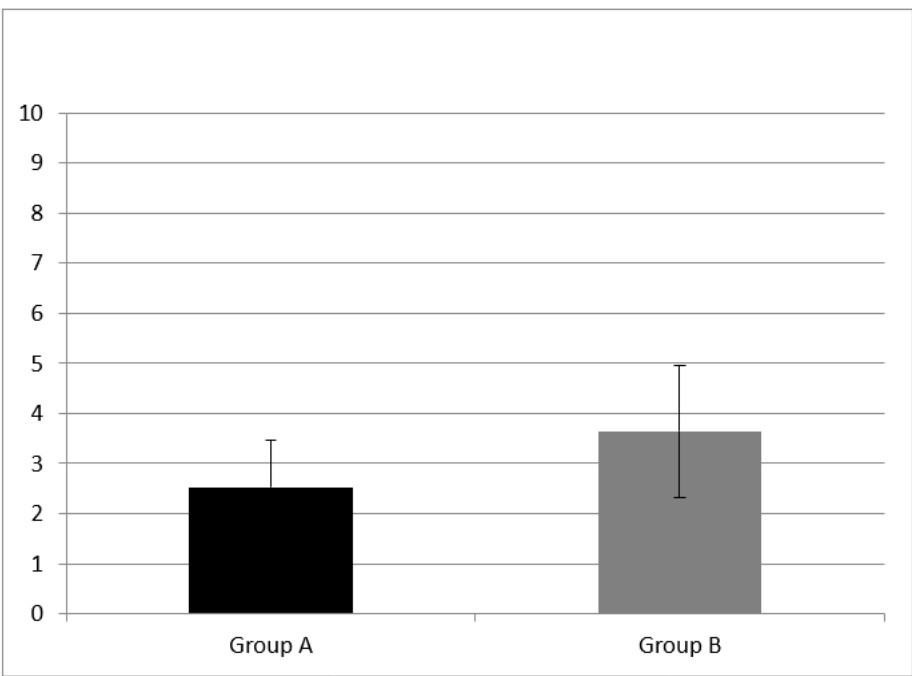

**Figure 2.** Working memory results at subtest of backward digit span in Group A and B. CI children (Group A) had backward digit span mean score of 2.53 ± 0.943; NH peers (Group B) had mean score of 3.64 ± 1.33. A statistical difference was found ($z = -2.30$, $p = 0.02$) Error bars indicate the standard deviation.

### 3.2. Inhibition and Control of the Impulsive Response-Caf

Group A presented inhibition and control of the impulsive response analyzed by CAF equal to 16.59 ± 5.84. Group B had a CAF result of 8.57 ± 8.89 (Figure 3). A statistically significant difference in CAF was found between Group A and B ($z = -2.767$, $p = 0.005$).

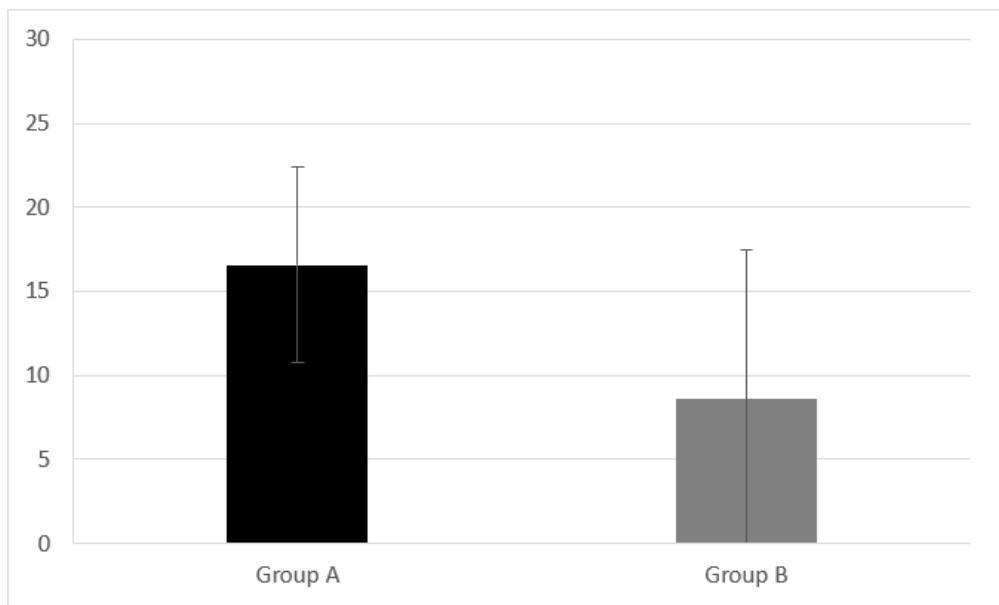

**Figure 3.** Inhibition and control of the impulsive response analyzed by CAF measured in Group A and B. CI children (Group A) had CAF equal to 16.59 ± 5.84. NH peers (Group B) had a CAF result of 8.57 ± 8.89. A statistical difference was found (z = −2.767, *p* = 0.005) Error bars indicate the standard deviation.

*3.3. Cognitive Flexibility*

Concerning phonemic fluency, the mean score was 14.00 ± 10.02 in children with CI (Group A) and 28.93 ± 13.53 in NH children (Group B). Finally, at categorical fluency, group A scored 34.00 ± 14 and Group B 57.07 + 18.42.

Statistically significant differences of phonemic fluency mean score (z = −3.02, *p* = 0.002) and categorical fluency mean score (z = −3.15, *p* = 0.001) between group A and B were found (see Figures 4 and 5).

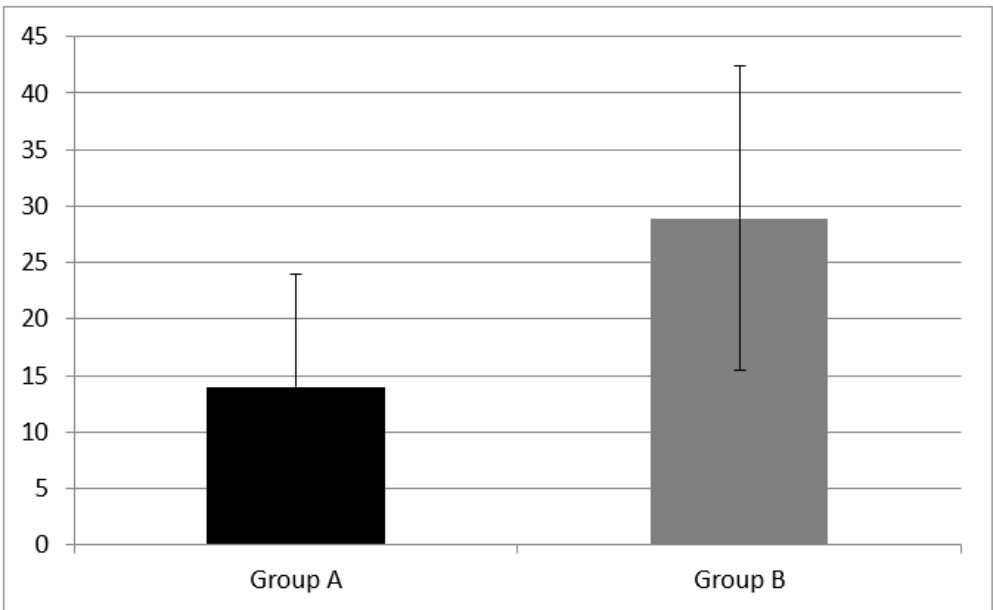

**Figure 4.** Cognitive flexibility skills measured as phonemic fluency in Group A and B. Phonemic fluency was equal to 14.00 ± 10.02 in CI children (Group A) and 28.93 ± 13.53 in NH children (Group B). A statistical difference was found (z = −3.02, *p* = 0.002). Error bars indicate the standard deviation.

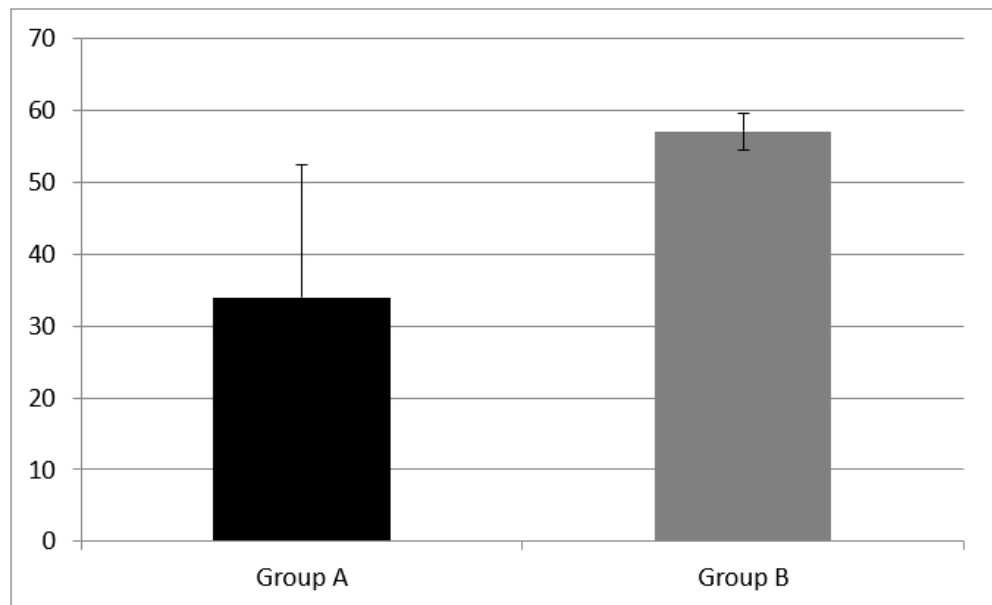

**Figure 5.** Cognitive flexibility skills measured as categorical fluency in Group A and B. Categorical fluency was equal to 34.00 ± 14 in CI children (Group A) and 57.07 + 18.42 in NH children (Group B). A statistical difference was found (z = −3.15, *p* = 0.001). Error bars indicate the standard deviation.

### 3.4. Planning

The TOL correct score was 22.59 ± 5.85 in Group A and 30.43 ± 2.53 in Group B.

A statistically significant difference of TOL total correct mean score (z = −3.71, *p* = 0.00) between group A and B was found (see Figure 6). In addition, there was a statistically significant difference in the TOL violated rules mean score (z = −4.3, *p* = 0.00) between groups A and B. CI children had a TOL violated rules mean score equal to 3.18 ± 2.96. NH children had 0 for TOL violated rules.

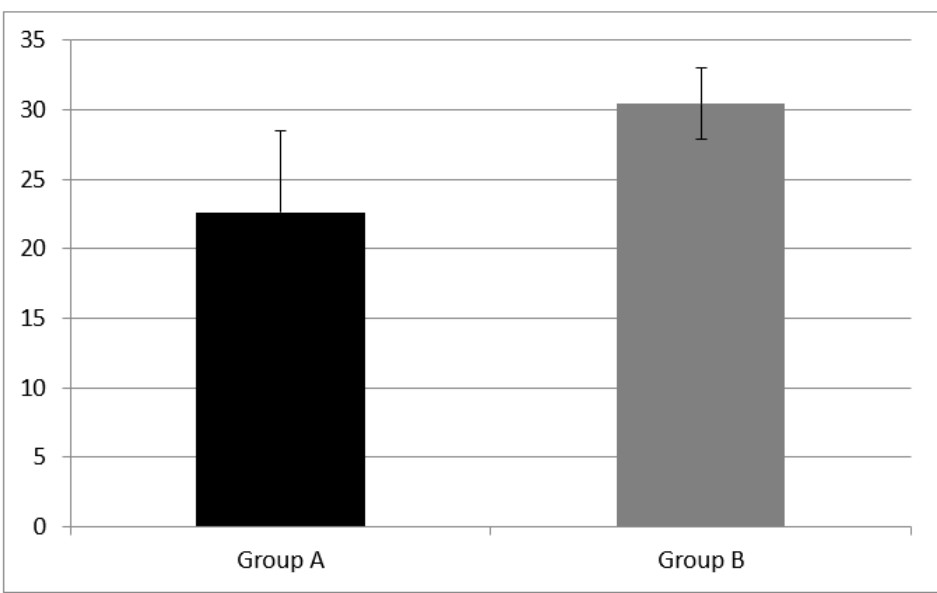

**Figure 6.** Planning results at subtest of TOL measured as total correct mean in Group A and B. TOL correct score was respectively 22.59 ± 5.85 in Group A and 30.43 ± 2.53 in Group B. A statistical difference was found (z = −3.71, *p* = 0.00). Error bars indicate the standard deviation.

### 3.5. Correlations between EFs and Age at CI Activation in the Group A

Statistically, significant direct correlations were found between age at CI activation and TOL total correct score (*p* = 0.003, rho: 0.671), backward digit span score (*p* = 0.025,

rho: 0.541), and categorical fluency score ($p = 0.042$, rho: 0.498). Statistically significant inverse correlation between age at CI activation and TOL violated rules score ($p = 0.015$, rho: $-0.579$) and TOL decision time score ($p = 0.034$, rho: $-0.532$) were found. In addition, there were statistically significant direct correlations between the forward digit span score and phonemic fluency ($p < 0.001$, rho: 0.755), backward digit span and categorical fluency ($p = 0.002$, rho: 0.695), and backward digit span and phonemic fluency ($p < 0.001$, rho: 0.786). These results suggest that age at CI activation could positively influence the WM tested as forward digit span, backward digit span, and planning skill explored by TOL. Consequently, WM capacity was strongly related to performance in other cognitive tasks such as categorical and phonemic fluency.

## 4. Discussion

The aim of this study was to evaluate the EF domains in a group of deaf children treated with CI compared to a sample of NH children. EFs were assessed using standardized tests. The hypothesis was that a long period of hearing deprivation affects EFs in addition to language considering the dynamic relationship that binds such elements. This hypothesis was supported because no children with CIs (group A) scored within the normal range in the tests administered for the evaluation of EFs domains. The same scores were significantly lower when compared with scores obtained by NH children (group B). They had better results in all EF domains studied. These results were similar to the data in previous literature [18–24]. The absence of auditory stimuli in the newborn and the slowdown in language development and brain plasticity resulting from the hearing deprivation period could be therefore considered potential alterations at the basis of the EF deficits in children with CIs.

The second aim was to analyze if the age at CI activation could positively influence EF development in group A. The obtained results showed that the age at CI activation resulted in better EF domain performances, specifically in working memory (backward digit span), planning (TOL), and cognitive flexibility (categorical fluency). These results are confirmed by numerous previous data and supported the important role of the unspecified connections between low-level sensation/perception and higher-order cognition (e.g., the auditory connectome model) [7,19,25].

In addition, in CI children involved in the study, better WM performances were correlated to better cognitive flexibility outcomes. These findings might be due to the fact that early recovery of auditory input favors the development of WM [26]. Furthermore, cognitive flexibility, which is the ability to shift attention from a particular task/mental state to another based on the analysis of the obtained/expected results, requires working memory skills [27,28].

The importance of exposure to sound as early as possible and of cochlear implantation performed at an early age is therefore deduced, considering that brain plasticity also acts within certain age ranges, with critical periods at 4 and 7 years old [29].

If the correction of hearing deprivation occurs before the age of 3.5, there is a recovery of auditory function, which is demonstrable by measuring the evoked auditory potentials. While there is a slower and partial recovery of auditory functions if hearing deprivation continues beyond 7 years of age (in our sample cochlear implantation was done at about 2 years of age) [30].

Some authors emphasized that healthy EF skills did not require audition and therefore that difficulties in this domain did not result primarily from a lack of auditory experience but from problems with language secondary to hearing loss [31,32]. Psycholinguistic studies have demonstrated the existence of a relationship between language and EFs identifying specific language aspects that require controlled processing, for example, resolving ambiguity by integrating context and inhibiting alternative meanings or overriding a regular past tense rule to correctly produce irregular verb forms. At a more practical level, this further theory has implications for clinical care: if auditory access is required for healthy cognitive development, then all deaf children need exposure to sound as early as possible.

However, if linguistic access is required, the space of possible interventions becomes bigger. There is no doubt that further studies using overlapping measurement methods are needed to confirm or refute one or the other hypothesis.

In our study, for children with pre-verbal profound deafness CIs, generating an improvement in hearing performance involved a long-term implementation of the examined EFs (planning, WM, cognitive flexibility) which, however, remained less efficient than for the NH children. It remains certain that early recovery of EFs in children with pre-verbal profound deafness should be a goal in a speech therapy rehabilitation after a CI, considering that EFs refers to a constellation of cognitive skills that regulate both cognition and behavior. If WM skills predict long-term language abilities, concerning receptive vocabulary acquisition and growth and word recognition [33], the early development of EF skills strongly predicts long-term outcomes including graduation, health, addiction, and socioeconomic status [34,35]. Consequently, EF development is closely connected with quality of life [36].

Some constraints limit the generalization of this study's results. The first limit was the size sample, which limited the statistical analysis. Therefore, the tests used for the evaluation of EFs create artificial situations and contexts that go beyond everyday situations.

The EF assessment was performed at a given time, school age, and no longitudinal data were available.

## 5. Conclusions

The results of this study highlight that cochlear implantation plays a role in improving hearing and consequently influences the development of language in CI children. This is a fundamental pre-requisite to support the development of EFs in deaf children. The evaluation of EFs domains should, therefore, be performed as routine tests of follow-up immediately after cochlear implantation to identify any difficulties early and intervene with an appropriate rehabilitation program intervention.

**Author Contributions:** All authors contributed equally. All authors have read and agreed to the published version of the manuscript.

**Funding:** This research received no external funding.

**Institutional Review Board Statement:** The study design and the subjects' recruitment were approved by the Institutional Review Board Statement of Hospital Consortium Policlinico of Bari, Italy, and were conducted according to the ethical standards of Hospital Consortium Policlinico of Bari, Italy. This research received the approval of the Ethics Committee of Hospital Consortium Policlinico of Bari, Italy (approval number: 3845, protocol code 264/CE).

**Informed Consent Statement:** Informed consent was obtained from all subjects families involved in the study.

**Data Availability Statement:** The data presented in this study are available on request from the corresponding author.

**Conflicts of Interest:** The authors declare no conflict of interest.

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
