# Peer review of "Executive Functions and Deafness: Results in a Group of Cochlear Implanted Children"

_audiolres, doi:10.3390/audiolres11040063_

Round 1

Reviewer 1 Report

I thank the Authors for this interesting manuscript and thanks to the Editor for allowing me to review it.

The importance of early neurocognitive rehabilitation and CI early hearing restoration in children with profound SSHL is well known, but this manuscript emphasizes that professional cooperation of Child Audiologists, Speech Therapists, and Child Neuropsychiatrists is mandatory to recognize and improve their executive functions.

Study design, patient selection criteria, test battery selection, statistical analysis, discussion, conclusion and references are relevant and pertinent, and add high scientific value to the content of the manuscript. For this reason, I think this manuscript should be published immediately.

I have just a few observations:

  • Please, specify all abbreviations the first time you use them;
  • Table I: please, check the English translation of the terms: "bilateral" and "bil";
  • Line 104: please, do the same for the statement: “Batteria di valutazione neuropsicologica per l’età evolutiva”;
  • Line 288: What do you mean by the term: “ecological”? Please, specify or delete it.

Thanks again to the Authors and to the Editor.

Best Regards

Author Response

Dear Reviewers,

first, many thanks for your revision, for having appreciate our paper and for your kind suggestions to improve it. We found all your comments to be valid and we modified the manuscript accordingly.

We reviewed the manuscript as suggested by reviewers point by point.

Reviewer #1.

  • Please, specify all abbreviations the first time you use them;
    • We specified all abbreviation the first time we used
  • Table I: please, check the English translation of the terms: "bilateral"and "bil";
    • We corrected the English translations
  • Line 104: please, do the same for the statement: “Batteria di valutazione neuropsicologica per l’età evolutiva”;
    • We translated the Test name
  • Line 288: What do you mean by the term: “ecological”? Please, specify or delete it.
    • We modified the sentence

Regards

The Authors

Submission Date 06/11/2021

Reviewer 2 Report

Thank you for the opportunity to review this study. I find this article very interesting, well structured, and comprehensive.

Have perception and language skills been assessed in CI users, in order to obtain a homogeneous group of children with a comparable CI performance? And what tools or scales have been used?

Has etiology of hearing loss, here not mentioned, been considered as a possible factor influencing the CI performance and therefore EF domains scores?

Do unilateral CI users included in this study use a bimodal stimulation? Have bilateral CI or bimodal stimulation been considered as a possible factor influencing EFs compared to unilateral CI?

Finally, I only suggest specifying the abbreviations throughout the abstract (e.g., EFs, CI, NH) and “WM” at line 252. In Table 1, “R”, “L”, and “bilateral” could be better explained (the reader cannot understand if R and L refer to the age of the first implant surgery and who are the 3 bilateral CI users). Table 2 was never mentioned in the text.

Furthermore, I think the Figure 4 to 7 do not show a correct vertical axis, because there is not correspondence with the results showed in text. Probably the caption of the Figure 5 is incorrect (perhaps it should be “categorical fluency” and not “phonemic fluency”).

Author Response

Dear Reviewers,

first, many thanks for your revision, for having appreciate our paper and for your kind suggestions to improve it. We found all your comments to be valid and we modified the manuscript accordingly.

We reviewed the manuscript as suggested by reviewers point by point.

Reviewer #2.

  • Have perception and language skills been assessed in CI users, in order to obtain a homogeneous group of children with a comparable CI performance? And what tools or scales have been used?
    • The perception and language skills were assessed with routine tests used in our clinic. The CI performance was homogeneus and we specified the Tests used in result section.
  • Has etiology of hearing loss, here not mentioned, been considered as a possible factor influencing the CI performance and therefore EF domains scores?
    • We specified the etiology of HL. The CI group was homogeneous and for this reason we did not consider the etiology as possible factor influencing EFs.
  • Do unilateral CI users included in this study use a bimodal stimulation? Have bilateral CI or bimodal stimulation been considered as a possible factor influencing EFs compared to unilateral CI?
    • Unilateral Ci used the bimodal stimulation. We had only 3 bilateral Ci patients. They had auditory and language outcomes like unilateral CI. For this reason and for the number of bilateral CI we decide to not consider as possible factor influencing EFs.
  • Finally, I only suggest specifying the abbreviations throughout the abstract (e.g., EFs, CI, NH) and “WM” at line 252. In Table 1, “R”, “L”, and “bilateral” could be better explained (the reader cannot understand if R and L refer to the age of the first implant surgery and who are the 3 bilateral CI users). Table 2 was never mentioned in the text.
    • We specified the abbreviations in all manuscript
  • Furthermore, I think the Figure 4 to 7 do not show a correct vertical axis, because there is not correspondence with the results showed in text. Probably the caption of the Figure 5 is incorrect (perhaps it should be “categorical fluency” and not “phonemic fluency”).
    • We corrected the figures

Regards

The Authors

Submission Date 06/11/2021

Reviewer 3 Report

This paper presents results for executive function (EF) assessments for 17 children using cochlear implants(CIs) and compares with a group of 14 normal hearing children of similar age range. The CI children were all congenitally profoundly deaf and were implanted at ages ranging from1 to 7 years.  We can assume that they had no useful auditory input prior to CI and thus are likely to have significant delays in spoken language development when implanted.  These delays would in turn be affected by a range of factors including age at implantation.  The main finding of the study is that the CI group performs significantly worse on all domains of EF that were tested.     This is to be expected, however, as language development is crucial to many of the EF domains, some of which directly assess language based ability.  I have a number of concerns about the paper as presented as I do not feel the concluding statements are really supported by the results.  The authors state in the conclusion that "cochlear implantation plays a role in determining  the development of EFs in deaf children". That may well be the case, and I hope that it is the case, but these results do not directly support this statement.  To know what the CIs are contributing we would have to know what the children's ability might have been if they did not have a cochlear implant (difficult), or compare two groups who received their implants, for example, before 2 years of age, and after 4 years of age.  There is a suggestion in the discussion that duration of CI use  resulted in better executive functions but I could not see the statistics that supported this in the results section.  There are results presented for age at CI but they need more explanation so that the reader can interpret what they mean. I feel the title is unnecessarily long.  The study is about EF in CI children and not really about the cooperation between the specialists involved.  It is great that a multidisciplinary approach was taken here and this should be highlighted in the text, but maybe it is unnecessary in the title. On additional important issue that should be addressed is that a CI is a device that can improve auditory input to the central auditory system.  By improving auditory input, the development of speech perception and spoken language can also be improved.  By improving language development, it is likely that EFs can improve.  The authors should be very careful about cause and effect in the discussion and conclusions.  A CI cannot directly affect EFs; it can only improve sensory input, that may improve one aspect of spoken language development, which in turn is integral to many aspects of EFs.  My specific comments are below.

p.1, l.13  need to identify abbreviation in first appearance

p.1, l.22  this statement implies causality - but these are only associations  

p.1, l.24  I question this statement.  One would have to argue that it is communication development that plays this role.  A cochlear implant may help a deaf child to improve their hearing which may in turn help their communication and cognitive development

p.1, l.29  Move "health" after "worldwide"

p.1, l.31  Vague.  This really means significant bilateral permanent (sensorineural) hearing loss

p.1, l.36  auditory rather than acoustic

p. 2, l.66 spelling for "Institutional"

p. 2, l.74  Change to "32 subjects were enrolled"

p.2, l.77  should be "consisted"

p.2, l.83  the table shows an average use time of 6.17 years.  Many subjects have less than 6 years. Do you mean 6 months?

Table 1  The number of significant figures for means, etc. unnecessary

Table 2  Categorical spelt wrong.  I prefer p<0.001 for the highly significant results

Figures 1 to 7  These bar graphs are not really adequate for describing the data.  The error bars are not explained (are they SD or confidence intervals).  Box and whisker plots would be better representation of the data or perhaps interval plots of means with confidence intervals. Figure 5 is particularly confusing - is the mean for group B zero? The scales are not labelled and the legends are not sufficient for the figures to be self explanatory.  It may be worth considering using non-parametric statistics throughout as the groups are quite small.

p.8, l. 226  There needs to be more explanation of the direction of these effects

p.9, l.228 spelling of Phonemic

p.9, l.235  "was supported" would be preferable to "proved correct"

p.9, 238-240 Sentence needs rewrite.  Next sentence also (l.240-241)

p.9, l.246  This is not the same as age at CI.  It is age at CI that is analysed in the results section.  This is more complex than is acknowledged by the authors.  There are three (related) time measures involved: age at CI, duration of CI use, and actual age at test.  Unravelling the effects of all of these is simply not possible in this small group

p.9, l.252  How was this determined?  The scores are correlated but the causality may be the other way or they could be both related to some other factor

p.9, l.254  sentence needs rewording - hard to follow

p.9, l.261  why 4 and 7 years - what is the evidence for this?

p.9, l.272-273  what is meant here? Is this a reference to use of sign language?

p.9, l.275-277  This sentence needs to be rewritten.  It is hard to follow

p.10, l.288  "impure" word choice?

p.10, l.293-294  Conclusion should change as it is not directly supported by the data. It is the pre-verbal hearing loss that has the detrimental affect on EF development due (probably) to delays in language development.  A CI may improve hearing early enough to change the trajectory of language development which should also help EFs.

p.10, l.297  There is no author contribution statement

p.10, l.306  There is no data availability statement

Author Response

Letter to the Reviewers

Dear Reviewers,

first, many thanks for your revision, for having appreciate our paper and for your kind suggestions to improve it. We found all your comments to be valid and we modified the manuscript accordingly.

We reviewed the manuscript as suggested by reviewers point by point.

Reviewer #3.

  • 1, l.13  need to identify abbreviation in first appearance
    • We specified the abbreviations in first appearance
  • 1, l.22  this statement implies causality - but these are only associations  
    • we modified the phrase
  • 1, l.24  I question this statement.  One would have to argue that it is communication development that plays this role.  A cochlear implant may help a deaf child to improve their hearing which may in turn help their communication and cognitive development
    • we modified the phrase
  • 1, l.29  Move "health" after "worldwide"
    • we moved "health" after "worldwide"
  • 1, l.31  Vague.  This really means significant bilateral permanent (sensorineural) hearing loss
    • We modified based on your suggestion
  • 1, l.36  auditory rather than acoustic
    • We modified based on your suggestion
  • 2, l.66 spelling for "Institutional"
    • we corrected the spelling
  • 2, l.74  Change to "32 subjects were enrolled"
    • We changed to  "32 subjects were enrolled"
  • 2, l.77  should be "consisted"
    • We corrected it
  • 2, l.83  the table shows an average use time of 6.17 years.  Many subjects have less than 6 years. Do you mean 6 months?
    • We corrected the mean
  • Table 1  The number of significant figures for means, etc. unnecessary
    • We apologize but we did not understand what the reviewer meant
  • Table 2  Categorical spelt wrong.  I prefer p<0.001 for the highly significant results
    • We corrected it
  • Figures 1 to 7  These bar graphs are not really adequate for describing the data.  The error bars are not explained (are they SD or confidence intervals).  Box and whisker plots would be better representation of the data or perhaps interval plots of means with confidence intervals. Figure 5 is particularly confusing - is the mean for group B zero? The scales are not labelled and the legends are not sufficient for the figures to be self explanatory.  It may be worth considering using non-parametric statistics throughout as the groups are quite small.
    • We modified the figures and legends
  • 8, l. 226  There needs to be more explanation of the direction of these effects
    • We explain better the result
  • 9, l.228 spelling of Phonemic
    • We corrected it
  • 9, l.235  "was supported" would be preferable to "proved correct"
    • We corrected it
  • 9, 238-240 Sentence needs rewrite.  Next sentence also (l.240-241)
    • We rewrote the sentence
  • 9, l.246  This is not the same as age at CI.  It is age at CI that is analysed in the results section.  This is more complex than is acknowledged by the authors.  There are three (related) time measures involved: age at CI, duration of CI use, and actual age at test.  Unravelling the effects of all of these is simply not possible in this small group
    • we corrected it
  • 9, l.252  How was this determined?  The scores are correlated but the causality may be the other way or they could be both related to some other factor
    • We rewrote the sentence
  • 9, l.254  sentence needs rewording - hard to follow
    • We modified the sentence
  • 9, l.261  why 4 and 7 years - what is the evidence for this?
    • We inserted the reference
  • 9, l.272-273  what is meant here? Is this a reference to use of sign language?
    • We explained better and rewrote the sentence
  • 9, l.275-277  This sentence needs to be rewritten.  It is hard to follow
    • We modified the sentence
  • 10, l.288  "impure" word choice?
    • We modified the sentence
  • 10, l.293-294  Conclusion should change as it is not directly supported by the data. It is the pre-verbal hearing loss that has the detrimental affect on EF development due (probably) to delays in language development.  A CI may improve hearing early enough to change the trajectory of language development which should also help EFs.
    • We changed it
  • 10, l.297  There is no author contribution statement
    • We completed it
  • 10, l.306  There is no data availability statement
    • We completed it

Regards

The Authors

Submission Date 06/11/2021

Round 2

Reviewer 3 Report

Thank you for taking my review comments into account in the revised manuscript.  I feel that my main concerns have been addressed.  There are just a few small corrections that are still needed.

p.1, l.15  Delete "divided"

p.1, l.18  spelling - "Neuropsychological"

p.1, l.22 delete "Sample's"

p.4, l.110 spelling

p.5, l.178  change "of" to "than"

p.9, l.264  change "positive" to "positively"

p.9, l.276  Delete "Similar"

p.9, l.290  "imput" should be "input"

p.10, l.297  "complete" implies that auditory function return to normal.  There is recovery of the cortical auditory evoked response, but this is not the same as "complete recovery of auditory function"  I suggest deleting "complete and rapid"

Author Response

Dear Reviewer,

first, many thanks for your revision. We found all your comments to be valid and we modified the manuscript accordingly.

We reviewed the manuscript as suggested by reviewers point by point.

Reviewer #3.

p.1, l.15  Delete "divided"

            We deleted it

p.1, l.18  spelling - "Neuropsychological"

            We corrected it

p.1, l.22 delete "Sample's"

            We deleted it

p.4, l.110 spelling

            We corrected it

p.5, l.178  change "of" to "than"

            We changed “of” to “Than”

p.9, l.264  change "positive" to "positively"

            We changed "positive" to "positively"

p.9, l.276  Delete "Similar"

We deleted it

p.9, l.290  "imput" should be "input"

            We corrected it

p.10, l.297  "complete" implies that auditory function return to normal.  There is recovery of the cortical auditory evoked response, but this is not the same as "complete recovery of auditory function"  I suggest deleting "complete and rapid"

            We deleted it

Regards

The Authors

Submission Date 11/11/2021